# Effect of Phytochrome Deficiency on Photosynthesis, Light-Related Genes Expression and Flavonoid Accumulation in *Solanum lycopersicum* under Red and Blue Light

**DOI:** 10.3390/cells11213437

**Published:** 2022-10-31

**Authors:** Pavel Pashkovskiy, Mikhail Vereshchagin, Vladimir Kreslavski, Yury Ivanov, Tamara Kumachova, Andrey Ryabchenko, Alexander Voronkov, Anatoliy Kosobryukhov, Vladimir Kuznetsov, Suleyman I. Allakhverdiev

**Affiliations:** 1K.A. Timiryazev Institute of Plant Physiology, Russian Academy of Sciences, Botanicheskaya Street 35, 127276 Moscow, Russia; 2Institute of Basic Biological Problems, Russian Academy of Sciences, Institutskaya Street 2, Pushchino, 142290 Moscow, Russia; 3Department Plant Physiology, Russian State Agrarian University—Moscow Timiryazev Agricultural Academy, Timiryazevskaya Street 49, 127550 Moscow, Russia; 4Tsitsin Main Botanical Garden, Russian Academy of Sciences, Botanicheskaya Street 4, 127276 Moscow, Russia

**Keywords:** BBX blue light, flavonoids, HY5, photosynthesis, red light, *Solanum lycopersicum*, transcription factors

## Abstract

The effect of red (RL, 660 nm) and blue (BL, 450 nm) light on *phy* mutant tomato plants was studied. The rates of photosynthesis (Pn) and transpiration, the efficiency of the primary photochemical processes of photosynthesis, the contents of flavonoids and phenolic compounds, the low-molecular-weight antioxidant capacity (Trolox equivalent antioxidant capacity (TEAC)) of leaf extracts, and the expression of light-dependent genes were evaluated. Under RL, BL, and white fluorescent light (WFL), the Pn values decreased in the order: WT > *phyb2* > *phyaphyb2* > *phyaphyb1phyb2*, except for the Pn in *phyb2* on BL. *Phyb2* also had a larger number of stomata under BL and, as a result, it reached maximum transpiration. The noticeable accumulation of flavonoids and phenolic compounds was observed only in the *phyb2* and *phyaphyb2* mutants upon irradiation with BL, which agrees with the increased TEAC in the leaf extracts. We suggest that the increased antioxidant activity under PHYB2 deficiency and the maintenance of high photosynthesis under BL are based on an increase in the expression of the early signaling transcription factors genes *BBX*, *HY5*. The largest decrease in the content of flavonoids and TEAC was manifested with a deficiency in PHYB1, which is probably the key to maintaining the antioxidant status in BL plants.

## 1. Introduction

Light controls many aspects of plant development through a complex signaling cascade, which, in addition to the photoreceptors, involves transcription factors (TFs), kinases, calmodulin, reactive oxygen species (ROS), etc. Light ensures the differentiation of chloroplasts under changing conditions, which makes it possible to regulate photosynthetic activity [1]. The phytochromes (PHYs) play one of the key roles in these processes. The phytochromes are red light (RL) and far-red light (FRL) sensors, and they regulate many developmental processes, including seed germination and hypocotyl growth. At least five types of phytochrome have been well identified; among them, the main ones are phytochrome A (PHYA) and phytochrome B (PHYB). PHYA is considered as the primary photoreceptor in the response to FRL, while PHYB plays the main role in the response to RL. The signal from phytochromes is transmitted either through various molecules (second messengers such as Ca^2+^, cAMP etc.) and/or through certain TFs to which photoreceptors bind in their active form [1,2]. The PHYs are inactive in the cytoplasm under the dark condition, and when exposed to RL, a change in the chromophore leads to rearrangements in the structure of apoproteins, which leads to their translocation to the nucleus [2,3]. In the nucleus, PHYs promote the degradation of phytochrome-interacting factors (PIFs), which suppress other positive photomorphogenesis regulators such as TFs HY5 and Golden2-like (GLKs) [4]. GLKs are required for the differentiation and maintenance chloroplast activity [3,5], while in *A. thaliana*, they are positively regulated by HY5 [6]. Other well-known receptors, the cryptochromes (CRY), perceive light in the blue and UV ranges of the spectrum. They are involved in the growth processes, de-etiolation of seedlings and circadian rhythms. They include two major cryptochromes (CRY1, CRY2) [7].

Changes in the phytochrome content significantly affect the content and activity of various components of phytochrome signaling as well as the genes for antioxidant enzymes and enzymes used for the biosynthesis of low molecular weight antioxidants and photosynthetic proteins [8]. This shift results in changes in the plant antioxidant status, photosynthesis efficiency, and other metabolic responses [8]. One of the main components of phytochrome signaling is TFs, which are involved in regulating most light-dependent genes. Zinc finger TFs constitute one of the most important families of transcription regulators in plants and play a central role in regulating plant growth and development under normal and stress conditions [9,10]. Among these TFs are B-box domain-containing proteins (BBX) that mediate their effect through interactions with components of the light signaling network, including TFs HY5, HYH and PIFs as well as the ubiquitin ligase COP1 [11]. For example, *AtBBX21* and *AtBBX22* contribute to the accumulation of transcripts for *HY5*, which is a key component of phytochrome signaling and can be labeled for proteasomal degradation via COP1-mediated ubiquitination [12]. In contrast, AtBBX24 and AtBBX25 suppress light signaling through interactions with HYH and HY5 [13]. Interestingly, AtBBX28 has been characterized as a light-induced repressor, because it could downregulate the transcriptional activity of HY5, allowing degradation in the dark via COP1 [14]. However, a signaling cascade of the TFs PIF3 and PIF1 has been shown to regulate the transcription of *AtBBX23*, the product of which interacts with HY5 to induce photomorphogenesis in *A. thaliana* seedlings [10,15]. These studies show that numerous BBX proteins, along with COP1 and HY5, play a critical role in light-dependent plant development. Additionally, the mechanisms of regulation for photomorphogenesis and the role of BBX in these processes have not been sufficiently studied. The tomato is a major vegetable crop that has a wide range of photoreceptor mutants, and these mutants are widely used in experiments to study the effect of light regulation on various physiological processes, including photosynthesis and light signaling [16].

The aim of this research was to study the role of various phytochromes in light signaling in tomato plants as well as to elucidate the extent to which this signaling determines the formation of flavonoids and other low-molecular weight antioxidants and affects photosynthesis in tomato leaves. We hypothesized that one of the possible reasons for the effect of phytochrome deficiency under the action of blue (BL) and red light (RL) is an increase in the expression of early response genes for TFs—*BBX* and *HY5* and a number of other light-dependent genes*,* which ultimately leads to the observed phenomena.

## 2. Materials and Methods

### 2.1. Plant Materials and Experimental Design

In the experiments, wild-type (WT) *Solanum lycopersicum* L. plants (Moneymaker cultivar, LA2706) and photoreceptor mutants *phyb2* (LA4358), *phyaphyb2* (LA4362), and *phyaphyb1phyb2* (LA4366) were used. The seeds were obtained from the Tomato Genetics Resource Center (TGRC) (University of California, Davis, CA, USA). The plants were grown for 30 days in a thermostatically controlled chamber with a 12 h photoperiod at a temperature of 23 ± 1 °C during the day and 23 ± 1 °C at night. Then, the plants were cloned using cuttings and grown for 2 weeks up to the age of 45 days. The plants were grown under white fluorescent lamps (WFL) (Philips, Poland) at a light intensity of 250 μmol photons m^−2^ s^−1^ in 8 × 8 × 10 cm vessels filled with perlite. Throughout the cultivation season, the plants were watered with a 2-fold diluted Hoagland nutrient solution. The experimental samples were irradiated for 0 d, 1 d and 7 d, 250 ± 15 μmol photons m^−2^ s^−1^ with red light (660 nm) and blue light (450 nm) LEDs (Epistar, Taiwan) (Appendix A). The spectral characteristics of the light sources were determined using an AvaSpecULS2048CL-EVO spectrometer (Avantes, The Netherlands). Plants used for the analysis of gene expression were sampled at 0 d, 1 d and 7 d of irradiation. All the remaining analyses, including microscopy, were produced on d 7 of the experiment. Six to ten of the most developed leaves from the second and third tiers were used for the analysis. Over the course of irradiation with light, determinations of the PSII activity and the intensity of photosynthesis and transpiration were performed on the leaves of intact plants; samples were taken simultaneously for microscopic analyses.

### 2.2. Analysis of Low Molecular Weight Antioxidants

The low-molecular-weight antioxidants were extracted with 80% methanol from leaves ground in liquid nitrogen.

The low-molecular-weight antioxidant capacity (Trolox equivalent antioxidant capacity (TEAC)) was determined spectrophotometrically according to the method described by Re et al. [17] involving the reaction of methanolic extracts with 2,2′-azino-bis [3-ethylbenzothiazoline-6-sulfonic acid] diammonium salt (ABTS) (Sigma-Aldrich, Burlington, MA, USA, CAS number 30931-67-0).

The total phenolics were determined spectrophotometrically using Folin–Ciocalteu phenol reagent (Sigma-Aldrich, Burlington, MA, USA; MDL number MFCD00132625) according to the procedure described by Singleton and Rossi [18]. The total phenolic content was expressed as gallic acid equivalents (GAE) in milligrams per gram of fresh weight (FW).

The total flavonoids were measured according to the methods of Kim et al. [19]. Afterwards, 1000 µL of distilled water, 150 µL of extracted sample and 50 µL of 5% NaNO_2_ were mixed together. After 6 min, 50 μL of 10% AlCl_3_ was added, and after another 5 min, 300 μL of 1 M NaOH was added to the mixture. The reaction mixture was homogenized, and after 10 min, the absorbance was measured at 510 nm. The total flavonoids were calculated by constructing a calibration curve using (+)-catechin hydrate (Sigma-Aldrich, Burlington, MA, USA, CAS Number 225937-10-0) and were expressed as milligrams of (+)-catechin per gram of FW [20].

### 2.3. Measurements of CO_2_ Gas Exchange and Transpiration

The photosynthetic rate (P_n_) and transpiration rate (T_r_) were determined in a closed system under light conditions using an LCPro + portable infrared gas analyzer from ADC BioScientific Ltd. (Hoddesdon, UK) that was connected to a leaf chamber with an area of 6.25 cm^2^. The CO_2_ uptake per leaf area (μmol m^−2^ s^−1^) was determined. The rate of photosynthesis of the leaves in the second layer from the top was determined at a saturating light intensity of 1000 μmol photons m^−2^ s^−1^. After the photosynthesis rate was measured, the light was turned off, and the rate of dark respiration was measured. The measurements were performed at 1000 μmol m^−2^ s^−1^ as well as before irradiation.

### 2.4. Determination of Photochemical Activity

The fluorescent induction curves were measured with a mini-PAM II fluorometer (Walz, Effeltrich, Germany) on plants adapted to the dark (30 min) as described earlier [21]. After a pulse of saturating light, the leaves of plants adapted to 30 min in the dark were kept in the dark for one minute, and then they were exposed to actinic light for 5 minutes, which was followed by saturating light pulses during which the parameters were measured. Blue LEDs (474 nm) were used to provide the measuring light (0.5 μmol photons m^−2^ s^−1^), actinic light (150 μmol photons m^−2^ s^−1^) and saturating pulses (474 nm, 5000 μmol photons m^−2^ s^−1^ and 800 ms duration). The parameter calculations on the basis of fluorescence data were performed using WinControl-3 v3.32 software (Walz, Effeltrich, Germany), and the formulas are taken from [22]. The values for the F_0_, F_v_, F_m_, F_m_′ and F_0_′, as well as the PSII maximum (F_v_/F_m_) and effective Y(II) (F_m_′ − F_t_)/F_m_′ photochemical quantum yields and nonphotochemical quenching (NPQ) (F_m_/F_m_′ − 1), were determined. F_m_ and F_m_′ are the maximum Chl fluorescence levels under dark- and light-adapted conditions, respectively. F_v_ is the photoinduced change in fluorescence, and F_t_ is the level of fluorescence before a saturation impulse is applied. F_0_ is the initial Chl fluorescence level. Actinic light was switched on for 10 min (I = 125 µmol photons m^−2^ s^−1^). The quenching parameters were also determined: NPQ—nonphotochemical fluorescence quenching; Y(NO)—quantum yield of nonregulated nonphotochemical energy dissipation in PSII and Y(NPQ)—quantum yield of regulated nonphotochemical energy dissipation in PSII, Y(NO) + Y(NPQ) + Y(II) = 1.

### 2.5. RNA Extraction and RT-PCR

RNA isolation was performed according to the TRIzol method (Sigma-Aldrich, Burlington, MA, USA). The quantity and quality of the total RNA were determined using a NanoDrop 2000 spectrophotometer (Thermo Fisher Scientific, Waltham, MA, USA). cDNA synthesis was performed using an M-MLV Reverse Transcriptase Kit (Fermentas, Waltham, MA, USA), an oligo (dT) 21 primer for nuclear encoding genes and a Random6 universal primer for chloroplast genes. The expression patterns of the genes were assessed using a CFX96 Touch™ Real-Time PCR Detection System (Bio-Rad, Hercules, CA, USA). Gene-specific primers (Appendix A) for anthocyanidin synthase (*ANS* NM_001374394.1), phytoene synthase (*PSY* NM_001247883.2), chalcone synthase (*CHS* NM_001247104.2), transcription factor elongated hypocotyl 5 (*HY5* NM_001247891.2), transcription factor phytochrome-interacting factor 4 (*PIF4* NM_001308008.1), E3 ubiquitin–protein ligase (*COP1* NM_001247118.2), phenylalanine ammonia-lyase 1 (*PAL1* XM_004249510.4), chloroplastic photosystem II nonphotochemical quenching protein (*PSBS* NM_001309257.1), de-etiolated1 (*DET1* NM_001247219.2), phytochrome-interacting factor 1b (PIF1b Solyc06g008030), phytochrome-interacting factor 3 (*PIF3* Solyc01g102300), phytochrome-interacting factor 7b (*PIF7* Solyc06g069600) (primers taken from [23]); photosystem II protein D1(*PSBA* Q2MIC0), photosystem II D2 protein (*PSBD* A0A0C5CUN5), chlorophyll *ab* binding protein 6A (*CAB6* P12360), phytochromobilin synthase (*Aurea* Q588D6), golden 2-like 1 transcription factor (*GLK1*, I6QQX8), B-box domain containing proteins (*BBX13* Solyc04g007210), B-box domain containing proteins (*BBX15* Solyc05g009310), B-box domain containing proteins (*BBX16* Solyc12g005750), B-box domain containing proteins (*BBX17* Solyc07g052620), B-box domain containing proteins (*BBX7* Solyc12g006240), B-box domain containing proteins (*BBX20* Solyc12g089240), B -box domain containing proteins (*BBX21* Solyc04g081020) and B-box domain containing proteins (*BBX30* Solyc06g063280) (primers taken from Bu et al., 2021) were selected using nucleotide sequences from the National Center for Biotechnology Information (NCBI) database (www.ncbi.nlm.nih.gov, accessed on 1 March 2022), https://www.uniprot.org/ accessed on 1 March 2022, https://phytozome-next.jgi.doe.gov/ accessed on 1 March 2022, with Vector NTI Suite 9 software (Invitrogen, Waltham, MA USA). The transcript levels were normalized to the expression of the *Actin1* gene. The gene expression in the wild type was given a value of 1. Changes in expression were considered significant with an increase or decrease in expression by at least 2 times relative to the WT control.

### 2.6. Scanning Electron Microscopy

The fragments (1 cm^2^) of fresh leaves from the middle part and the edge of the leaf blade were set on 2 cm × 4 cm copper plates. To obtain greater detail of the microstructure at high magnifications, the samples were frozen on a massive metal holder at −20 °C. The plate with a fresh sample was then fixed on the cooling stage of the Deben Coolstage refrigerating unit (UK) at −30 °C. The samples were imaged by a LEO-1430 VP (Carl Zeiss, Berlin, Germany) scanning electron microscope in high-vacuum mode operating at 20 kV with a backscattered electron detector QBSD and a working distance of 8–12 mm (cryoSEM).

### 2.7. Statistics

The fluorescence and CO_2_ gas exchange measurements were performed in four biological replicates. The scanning electron microscopy was performed in six biological replicates. Each plant sample fixed in liquid nitrogen was treated as a biological replicate; therefore, there were three biological replicates for the determination of Trolox equivalent antioxidant capacity, total phenol and flavonoids content as well as for gene expression analyses. For each of these experiments, at least three parallel independent measurements were performed. The significance of the differences among the groups was calculated by one-way analysis of variance (ANOVA) followed by Duncan’s method using SigmaPlot 12.3 (Systat Software Inc., San Jose, CA, USA). Letters indicate significant differences between the WT and the mutants (*p* < 0.05). The data are shown as the arithmetic means ± standard errors.

## 3. Results

### 3.1. Low-Molecular Weight Antioxidant Capacity

The highest antioxidant activity was observed in the *phyb2* and *phyaphyb2* mutants under WFL, while in the triple mutant and WT, this indicator was lower. Under RL, the antioxidant activity was the lowest in *phyb2* among all the variants, and the highest value was in the double mutant. The impact of BL was significantly different in terms of nonenzymatic antioxidant activity in all the studied samples. The activity of antioxidants increased in all the variants; however, the highest value was in the *phyb2* mutant (50.5), which was 2.6 times higher than in the WT under these conditions, and for the *phyb2* mutant itself, it was the highest among the studied types of light (Table 1).

### 3.2. Total Phenolics Content

When growing plants on WFL, the highest content of phenolic compounds was observed in the *phyb2* (2.1 times higher than control) and *phyaphyb2* (41.5% higher than control) mutants, while there were no differences between the WT and the triple mutant. Under RL, *phyb2* had the lowest (compared to WT) content of phenolic compounds (0.69), while the double mutant had the highest content (1.3). The effect of BL was different in that in all the variants, the content of phenolic compounds increased relative to other types of light; however, the largest increase was in the *phyb2* (2.5 times) and *phyaphyb2* (2.1 times relative to control) mutants (Table 1).

### 3.3. Flavonoid Content

The increase in flavonoid content in comparison with the control was also observed in the *phyb2* (by 65.9%) and double (by 34.1%) mutants under the WFL. RL caused an increase in flavonoids in the double mutant by an average of 72.9% relative to the other options. Exposure to BL resulted in a 2.8-fold increase in flavonoid content in *phyb2* and a 2.2-fold increase in *phyaphyb2* relative to the control and triple mutant (Table 1).

### 3.4. Measurements of CO_2_ Gas Exchange and Transpiration

When plants were irradiated with WFL and RL, the P_n_ values decreased in the order WT > *phyb2* > *phyaphyb2* > *phyaphyb1phyb2*. However, under BL conditions, the decrease in Pn changed somewhat: WT = *phyb2* > *phyaphyb2* > *phyaphyb1phyb2.*

The transpiration rates in the WFL variant were comparable for the WT and the double mutant 0.3–0.4 µM H_2_O m^−2^ s^−1^, while the rates for the triple and *phyb2* mutant were significantly lower (0.1–0.2). In *phyb2*, transpiration under RL was 4 times higher than that under WFL; in other variants, RL caused a decrease in transpiration, but the greatest decrease was observed in the WT and the triple mutant, and in *phyb2* and the double mutant, the values were comparable at 0.38–0.48. The greatest changes in transpiration were observed under BL, since in this case, transpiration was higher relative to other irradiation options. In addition, in the WT and the triple mutant, it was comparable (0.5), and its largest increase was in the *phyb2* mutant (2.5 times higher than the WT control) (Table 1).

### 3.5. Determination of Photochemical Activity

The maximum quantum yield of PSII remained stable in the studied plants on WFL and BL, and a decrease in this indicator was observed under RL; the largest decrease was in the triple mutant (0.794). The effective quantum yield was the highest for the WFL double mutant (0.50), and under BL, the *phyb2* mutant had the highest value (0.54), while the triple mutant had the lowest value of 0.34. RL caused a decrease in the Y(II) index, with the greatest decrease observed in the triple mutant (0.31); in other plants, the Y(II) index on BL was comparable. The nonphotochemical quenching indices were the lowest in the *phyb2* mutant under BL (NPQ 0.63; Y(NPQ) 0.17), while in other variants, this indicator was significantly higher; for example, the NPQ in the WT under BL exceeded 1. The highest values of the Y(NPQ) indices were in the triple mutant under RL (0.33) (Table 2).

### 3.6. Gene Expression

Initially, at the beginning of the experiment, the *phyb2* mutant showed increased expression of the *HY5* (8.3 times), *PSBS* (3.5 times), *ANS* (4.5 times), *CHS* (5.5 times), and *Aurea* (2.0 times) and *BBX7* (almost 3 times) genes relative to the WT control. The transcript levels of the other genes changed insignificantly. The double mutant initially had a high level of expression for the *PAL1* (by 9.2 times), *BBX15*, *BBX16*, *BBX20*, *BBX21*, *BBX30* (by 2.5–3 times), and *BBX7* (by 6.1 times) genes relative to the control WT. The transcript levels of other genes in the double mutant changed insignificantly. In the triple mutant, the expression of *HY5*, *PSBS*, *PSY*, *PAL1*, and *BBX16* increased seven to nine times relative to the WT control. Additionally, the triple mutant initially had a high level of *ANS*, *GLK1,* and *BBX30* expression (an average increase of 2.3 relative to the WT control) (Figure 1).

During the first 24 h of irradiation, RL caused an increase in *PIF3* expression in all the variants, in particular, in *phyb2* by 3.1 times, in *phyaphyb2* by 4.5 times, and in the triple mutant by 6.2 times relative to the WT. The expression of *PIF1* under RL increased (by 2.6 times) only in *phyb2* and did not change in the other mutants. The expression of *PIF4* under RL slightly increased in the double mutant but did not change in the other variants. The level of *HY5* transcripts under RL did not change in the triple mutant and increased by 3.5 times in *phyb 2* and 5.3 times in *phyaphyb2.*

The expression of *COP1* under RL increased by 2.7 times in *phyb2* and did not change in the other mutants. The expression level of the *Aurea* gene under RL increased by 2.1 times in the double mutant and by 3.4 times in the triple mutant, while it did not change in the *phyb2* mutant. The expression of *BBX13*, *BBX15*, and *BBX16* increased under RL in *phyb2* from 6.2 to 9.9 times. It is noteworthy that no increase in the expression of these genes was noted in the other mutants, with the exception of the triple mutant, in which the expression of the *BBX16* gene increased twofold relative to the WT. The level of *BBX17* transcripts decreased in all the variants under RL relative to the WT. The level of *BBX7* transcripts increased (3.8 times) only in the *phyb2* mutant under RL; simultaneously, the *BBX30* expression was 3.1 to 4.6 times higher than the control in all the variants. The highest expression of *BBX26* was observed in the triple mutant under RL. The expression levels of the study proteins PSII *PSBA*, *PSBD*, *CAB6*, and *PSBS* decreased in all the variants under RL, with the exception of a 3.0-fold increase in *PSBS* expression in *phyb2* relative to the WT. The expression of *ANS* increased under RL by 3.3 times in *phyb2* and the double mutant, while no changes in expression were noted in the triple mutant. The level of *PSY* transcripts increased under RL in all the variants, but the highest increase was in the triple mutant by more than nine times relative to the control. *CHS* expression increased 4.7 times only in the double mutant after 24 h of RL irradiation, and the expression level of the *PAL1* gene did not change in all the variants under RL (Figure 1).

Under BL irradiation on the first day of the experiment, *PIF1* expression increased by 8.9- and 7.9-fold in the double and triple mutants, respectively, but not in *phyb2*. The level of *PIF3* transcripts under BL increased in the double mutant by 2.3, and in the triple mutant, it increased by 9.85 times relative to the control. The expression level of *PIF7* was high in the *phyb* and *phyaphyb2* mutants, while it did not change in the triple mutant. The expression level of *PIF4* increased on the first day of BL irradiation in the double and triple mutants but decreased in *phyb2*. Under BL, *HY5* expression increased only in the *phyb2* mutant (by 8.9 times relative to the control). The expression of *DET1* after 24 h under BL increased 2.4-fold in *phyb2* and 4.5-fold in the triple mutant but decreased in the double mutant. The level of *COP1* transcripts increased 2.0-fold in *phyb2* and 3.5-fold in the *phyaphyb1phyb2* mutant, while it decreased in the double mutant. The *Aurea* expression did not change under BL during the first day. The level of *GLK1* transcripts increased under BL in the *phyb2* and triple mutants by an average of 4.4 times but did not change in the double mutant. The expression of all the studied TF *BBX* genes, with the exception of *BBX30*, significantly increased in the *phyb2* mutant. In the double mutant under BL, the expression of *BBX13*, *BBX7*, *BBX21* and *BBX25* increased, while the expression of the remaining *BBX* did not change. In addition, the expression of TF *BBX16* was observed in the triple mutant, which was 9.6 times higher than the control, and *BBX30* expression *was* almost 7 times higher than the control (8.7 times for *phyb2* and 25.8 times for *phyaphyb2*) (Figure 1).

Seven days after the start of the experiment, in the *phyb2* mutant under BL, the expression of the *PIF1*, *PIF3*, *PIF4*, *COP1*, *psbD*, and *BBX20* genes remained at a level two to three times higher than that in the control. The expression of the *CAB6, PSBS,* and *CHS* genes was four times higher than that in the control, and the expression of the *HY5* gene was 7.8 times higher. In the double mutant, after 7 days of exposure to BL, high expression was retained in the *psbA*, *PSBD*, *PSBS*, *BBX16* and *BBX30* genes (two to three times higher than WT), and the expression of *BBX17* was 5.2 times higher than that of the WT. In the triple mutant on day 7 of the experiment, only the expression of the *BBX16*, *BBX17*, *BBX26*, and *BBX30* genes remained three to nine times higher than that in the WT control. Under RL, after 7 days of the experiment, the *phyb2* mutant retained expression only in *PSBA*, *PSBD*, *PSBS*, *CHS, and ANS* (two to seven times higher than the WT control). In the double mutant, on the 7th day of the experiment under RL, expression was preserved in *PIF3*, *PSBS*, *PSBD*, *ANS*, and *BBX25* (from 3.5 to 9.7 times relative to the WT control). The triple mutant under RL retained the expression of *PSBA*, *PSBD*, *PSBS*, *PSY*, *BBX16*, and *BBX25* on day 7 of the experiment (at two to eight times higher relative to the WT control) (Figure 1).

### 3.7. Scanning Electron Microscopy

BL caused an increase in the number of trichomes of the first type in the *phyb2* mutant on the abaxial side of the leaves (Figure 2a and Figure 3f); along with this increase, the number of trichomes also increased in the triple mutant, but they were underdeveloped (Figure 2a and Figure 3h).

One of the features of the triple mutant was an increase in the number of epidermal cells on both sides of the leaf under BL and RL light (Figure 2c,d, Figure 3h,I and Figure 4h,i), while this increase was not observed under WFL (Figure 2c,d and Figure 4d).

Light affected both the number of stomata and their location. Under BL, all mutants showed an increase in the number of stomata on the abaxial side of the leaves (Figure 2e and Figure 3f,g,h). Moreover, RL caused a decrease in the number of stomata in the *phyb2* mutant (Figure 2e and Figure 3j), which was not observed in other plants. The triple mutant also had an increase in the number of stomata (Figure 2e), with many stomata being laid in the epidermis but remaining underdeveloped, especially under RL (Figure 3l). Another feature of the *phyb2* mutant was the formation of a large number of stomata on the adaxial side of the leaf under BL (Figure 2f).

RL on the double mutant caused papillose formations on the outer surface of the main cells of the epidermis, which are characteristic only of this mutant (Figure 4k).

In the triple mutant, BL and RL caused an increase in the number of secretory multicellular trichome types III, VI, and VII (Figure 4h,l), which was not observed in the other plants. In addition, a large number of underdeveloped trichomes were observed in the triple mutant (Figure 3h,l and Figure 4h,l). Other stomata were embedded in the epidermis, and their guard cells were filled with deposits of epicuticular wax (Figure 3l).

## 4. Discussion

The RL receptor phytochrome interacts with a large amount of TF and is able to regulate photosynthesis, the formation of chloroplasts and the synthesis of photosynthetic pigments as well as the formation of secondary metabolites in the leaves and fruits of plants [24]. These components include TF PIF, HY5, COP1, DET1, BBX, GLK1 and many others. For example, TF HY5 regulates the accumulation of anthocyanin by directly binding to the promoters of genes to biosynthesize these pigments [25]. DET1 deficiency under high light conditions affects the accumulation of flavonoids in tomato plants [26]. In this work, we evaluated the effect of phytochrome deficiency on photosynthesis and the accumulation of low molecular weight antioxidants and the expression of light-dependent genes in tomato leaves exposed to red and blue light. In particular, it was important to understand how phytochromes and related TFs can be involved in regulating the content of phenolic compounds, primarily flavonoids, in tomato plants under RL and BL.

It is known that changes in the phytochrome system and its signaling components lead to rapid regulation of the expression of genes encoding photosynthetic proteins and antioxidant enzymes, which affect photosynthetic activity and the antioxidant status, changing the content of low-molecular weight antioxidants and antioxidant enzyme activity [7,8]. In addition, phytochrome-induced effects depend on the light quality in which plants grow [27].

Upon activation by light, PHY phosphorylates the COP1 E3 ubiquitin ligase, leading to its inactivation. In the light, TF HY5 (which is negatively regulated by COP1 in the dark) activates the expression of genes associated with photomorphogenesis as well as the biosynthesis of some secondary metabolites [25]. In fact, in phytochrome-deficiency mutants, we observed a change in the flavonoid content (Table 1). Additionally, the greatest changes were observed under BL in *phyb2* and *phyaphyb2*. In addition, during the first 24 h under BL, *phyb2* showed an increase in *HY5* expression against the background of a decrease in the expression of *PIF3*, *PIF4* and *COP1*, which did not occur in other variants (Figure 1). It is likely that a decrease in the phytochrome signaling in the *phyb2* mutant under BL provides an enhancing of positive factors of photomorphogenesis, such as *HY5* and *GLK1*, which not only activates the flavonoids biosynthesis but also increased the transpiration rate (Table 1). In addition, the *phyb2* mutant under BL showed the maximum number of stomata, both on the lower and upper sides of the leaf, which also contributes to the activation of transpiration (Table 1; Figure 2, Figure 3 and Figure 4).

The observations made here are additionally confirmed by fluorescent parameters; in particular, the *phyb2* mutant under BL showed the highest effective quantum yield (Y(II)) and the lowest nonphotochemical quenching NPQ and Y(NPQ) (Table 2). All this was accompanied by an increase in the expression of the main genes (*psbA*, *psbD*, *CAB6* and *psbS*) of PSII proteins over the first 24 h of the experiment, which was not observed in all other variants (Figure 1). This increase probably indicates the plasticity of the photosynthetic apparatus of the *phyb2* mutant, which is higher than in the WT.

The phytochrome family of tomato consists of five genes, *PHYA*, *PHYC*, and three members of the *PHYB* subfamily, *PHYE* and two paralogs of *PHYB* that have arisen through independent duplication (*PHYB* and *PHYD* in *A. thaliana*, *PHYB1* and *PHYB2* in *S. lycopersicum*). It is likely that PHYC and PHYE can partly replace key PHYA, PHYB1 and PHYB2 in the synthesis of phenolic compounds; furthermore, a difference in the content of these compounds between WT and mutants was not substantial. Under BL, the role of all phytochromes was reduced, and a difference between WT and mutants was more significant. In addition, the photosynthetic proteins D1 (*psbA*) and D2 (*psbD*), whose gene expression is important for the light-induced recovery of PSII, and the antenna complex gene (*CAB6*), which is important for the functioning of the antenna Chl–protein complex, were minimum in a triple mutant grown under RL and especially BL. It is likely that this is one of the main reasons for the strong decrease in photosynthetic activity in a triple mutant with the deficiency of key phytochromes (PHYA, PHYB1 and PHYB2). We can see that PHYB1 plays a key role in this decrease. It is likely that phytochromes can affect photosynthesis by the control of photosynthetic proteins.

Apparently, TF *BBX* in the *phyb2* mutant can be a positive regulator of photomorphogenesis and it can be the gene involved in the primary response to changes in the quality of light, triggering the biosynthesis of flavonoids and activating photosynthesis and antioxidant potential in the *phyb2* mutant under BL. This observation is consistent with the fact that when plants are irradiated with TFs, BBX can bind to the promoter regions of *HY5* and activate its transcription [24]. Moreover, BBX21 and HY5 can bind to the promoters of other *BBX* genes and activate their transcription [28]. Thus, BBX and HY5 TFs form a regulatory cluster capable of controlling plant photomorphogenesis. In this study, we were able to detect an increase in the expression of several key BBXs under BL in the *phyb2* mutant. Thus, within the first 24 h of the experiment, the levels of *BBX13*, *BBX15, BBX16*, *BBX17*, *BBX7*, *BBX20*, *BBX21*, *BBX25*, and *BBX26* transcripts increased from two to nine times relative to the WT control (Figure 1). It should be noted that the expression of only one gene, *BBX30*, decreased in the *phyb2* mutant. In addition, the expression of *BBX30* increased 7-fold relative to the WT in the triple mutant (Figure 1). In addition, we observed an increase in the expression of *PAL1* under BL, and it is one of the main enzymes required for flavonoid synthesis (there was a 9-fold increase in the expression in *phyb2* relative to the WT) (Figure 1). We hypothesize that BBX, directly or indirectly through HY5, can upregulate *PAL1* expression under BL.

A question arises as to what can affect the activation of BBX itself under BL conditions when phytochrome signaling is reduced. Previously, it was shown that *BBX4*, *BBX23* and *BBX29* in tomato plants were only expressed in response to RL, while *BBX7*, *BBX13* and *BBX25* were expressed in response to FRL [29]. These results indicate that BBX proteins can directly interact with phytochrome photoreceptors. In addition, PHYB was found to directly interact with BBX and positively regulate its accumulation under RL in *A. thaliana* [30], indicating that photoreceptors can directly control some BBX. In our experiments, we were able to show that PHYB1 is a receptor that is involved in light signaling, which determines the biosynthesis of flavonoids under BL conditions, because the deficiency of these phytochromes in the triple mutant leads to a decrease in the amount of phenolic compounds and flavonoids (Table 1). It is also known that TF GLK may be involved in the PHY-mediated regulation of BBX, which is confirmed by the presence of binding domains in the promoter regions of the *BBX*, *HY5*, and *PIF* genes [10]. Interestingly, in our experiments, the highest expression of *GLK1* was observed in the triple mutant under RL against the background of increasing expression of *BBX21*, *BBX25*, *BBX26*, *BBX30*, and *PIF3* (Figure 1). However, the amount of flavonoids in this variant did not increase (Table 1). It can be assumed that *BBX21*, *BBX25*, *BBX26*, and BBX30 are weakly involved in flavonoid biogenesis in the triple mutant. This assumption is supported by the fact that *BBX25* expression was preserved by day 7 only in the triple mutant under RL and BL, and the contents of flavonoids was reduced (Figure 1, Table 1).

Most of the flavonoids contained in the tomato leaf are synthesized in special secretory cells called secretory trichomes [31]. The tomato has several types of secretory trichomes (types 1, 6 and 7), which are most often located on the abaxial side of the leaves. Confirming our results, under BL, the highest number of type 1 secretory trichomes was found in *phyb2* mutants on the abaxial side of the leaves (Figure 2). Additionally, a large number of secretory trichomes were found in the triple mutant under RL; however, most of them were underdeveloped and most likely not able to accumulate flavonoids (Figure 3). We suggest that trichomes in the *phyb2* mutant may be one of the possible sites for the accumulation of flavonoids. In addition, the double mutant under RL showed papillose formations of epidermal cells on the adaxial side of the leaves, which were not found in any other mutant (Figure 4). It can be assumed that this is one of the possible mechanisms for reducing the penetration of RL into the leaf mesophyll. The change in epidermal cell size in the double mutant could theoretically indicate the involvement of PHYA in cell elongation, which occurs due to a change in the ratio of the main phytohormones [32,33]. This assumption is supported by data on the number of epidermal cells. Thus, the highest number of cells on both the abaxial and adaxial sides of the leaves was found in the triple mutant, and their number was four times higher than that in the WT and other mutants. This result indicates a change in the hormonal status of the triple mutant and a possible increase in the level of cytokinins relative to auxins [34].

The content of flavonoids in our experiments correlates with the antioxidant capacity of leaf extracts (*r* = 0.99, data not shown). Before starting the main experiments, we performed a screening analysis of a large group of tomato light-receptor mutants under conditions of light of different spectral composition. The results obtained are presented in Appendix A. It was shown that blue light that had the greatest effect on the accumulation of flavonoids in leaves, and *phyb1* mutations led to a decrease in the flavonoids content. The highest nonenzymatic activity was observed in the *phyb2* mutant under BL (Table 1). *BBX* proteins also play a vital role in the regulatory networks that control plant adaptations to abiotic stress. Previous studies have shown that both BBX5 and BBX21 positively regulate plant tolerance to drought and salt stress in *A. thaliana* [35]. We hypothesize that the resistance of plants in previous studies is due precisely to the increased content of flavonoids and that *phyb2* plants grown on BL will also be more resistant to some abiotic stresses. Thus, PHYB1 and PHYB2 seem to play different roles under conditions of sufficiently high BL intensity. CRY1 plays an important role in the resistance of plants grown under BL to high-intensity light and UV-B [36]. It can be assumed that PHYB1 and PHYB2 in tomato plants are also important elements of this resistance.

We assume that the obtained results can be used to obtain new more resistant tomato varieties for outdoor cultivation. We also assume that it is possible to create conditions for the translocation of flavonoids from leaves into tomato fruits, which will allow creating products with increased nutritional value.

## 5. Conclusions

BL had the greatest effect on the accumulation of flavonoids and phenolic compounds in the *phyb2* mutant, which was confirmed by an increase in the TEAC of the leaves extracts. We suggest that the observed phenomena are based on the regulation of the expression of components phytochrome signaling genes including TF HY5 and BBX. This expression is markedly reduced in PHYB1 deficiency but increased upon PHYB2 deficiency under BL conditions. We believe that PHYB1 is a key phytochrome because its deficiency in the triple mutant leads to a maximum decrease, low-molecular weight antioxidant capacity represented by flavonoids and also in photosynthetic activity under BL.

## Figures and Tables

**Figure 1 cells-11-03437-f001:**
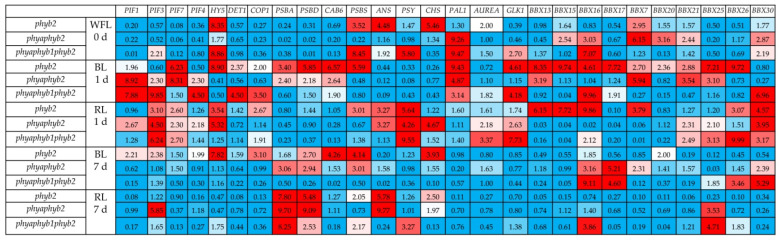
Influence of WFL, BL and RL on the levels of the anthocyanin synthase genes *ANS*, phytoene synthase *PSY*, chalcone synthase *CHS*, transcription factor elongated hypocotyl 5 *HY5*, transcription factor phytochrome-interacting factors *PIF1*, *PIF3*, and *PIF4*, E3 ubiquitin–protein ligase *COP1*, phenylalanine ammonia-lyase 1 *PAL1*, chloroplastic photosystem II nonphotochemical quenching protein *PSBS*, de-etiolated1 *DET1*, photosystem II protein D1 *psbA*, photosystem II D2 protein *psbD*, chlorophyll ab binding protein 6A *cab6*, phytochromobilin synthase *Aurea*, golden 2-like 1 transcription factor *GLK1*, B-box domain containing proteins *BBX13*, *BBX15*, *BBX16*, *BBX17*, *BBX7*, *BBX20, BBX21*, *BBX25*, *BBX26* and *BBX30* effect on leaves of mutant tomato plants on genes for the major phytochromes, WFL 0 d—starting point experiment, 1 d—1 day of light exposure, and 7 d—seven days of exposure to light. The transcript levels were normalized to the expression of the *Actin1* gene. The gene expression in the WT was used as one unit. Changes in expression were considered significant, with an increase (red light) or decrease (blue light) in expression by at least two times relative to the WT control.

**Figure 2 cells-11-03437-f002:**
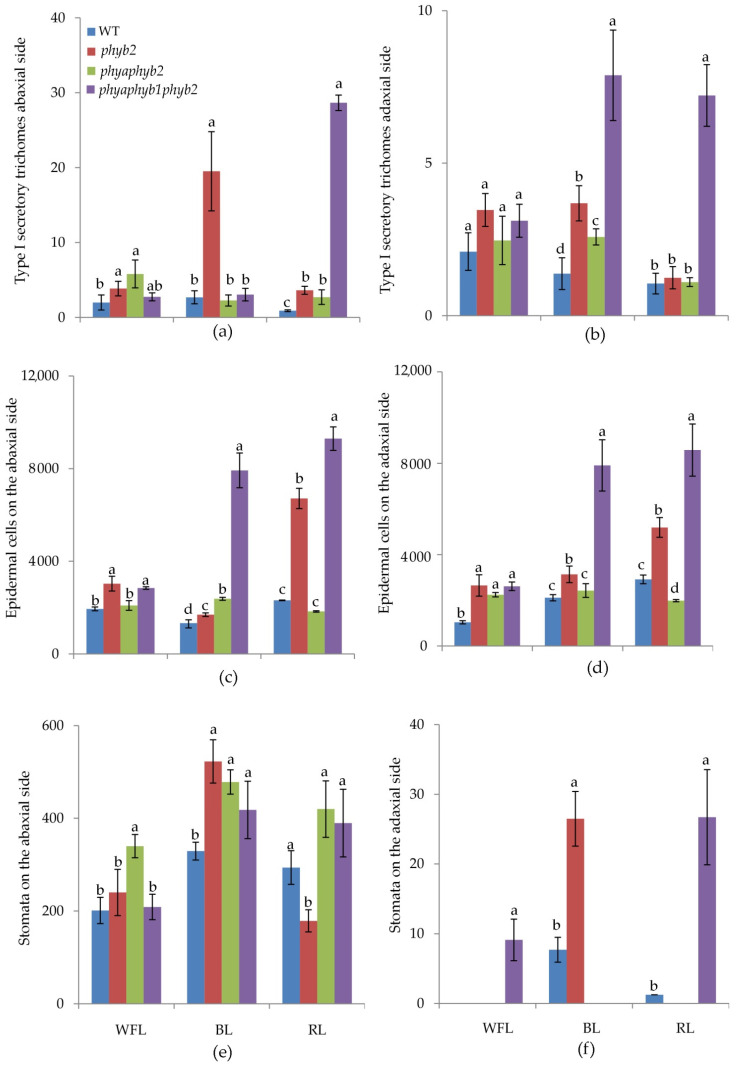
Effect of WFL, BL, and RL on the number of type I secretory trichomes (**a**,**b**); number of epidermal cells (**c**,**d**) and number of stomata (**e**,**f**) per mm^2^ on abaxial (**a**,**c**,**e**) and adaxial (**b**,**d**,**f**) sides of leaves. Different letters indicate significant differences (*p* < 0.05) between the experimental treatments. The means ± standard errors, *n* = 6.

**Figure 3 cells-11-03437-f003:**
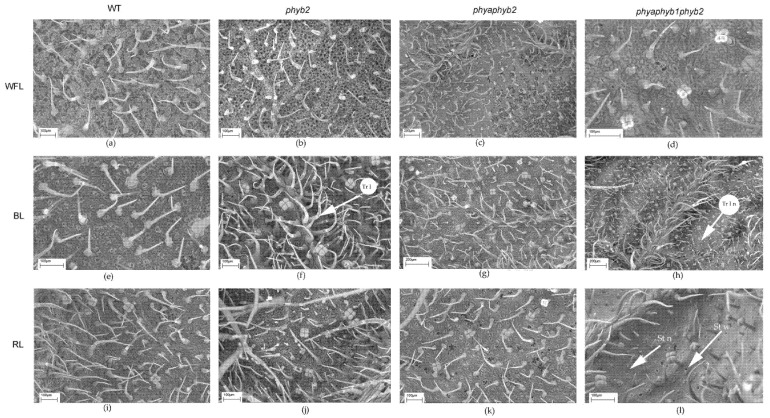
SEM photograph of the abaxial surface of tomato leaf mutants for the main phytochrome genes: WT (**a**,**e**,**i**); *phyb2* (**b**,**f**,**j**); *phybaphyb2* (**c**,**g**,**k**) and *phyaphyb1phyb2* (**d**,**h**,**l**) under the action of WFL (**a**–**d**), BL (**e**–**h**) and RL (**i**–**l**). Tr I—trichomes of type I; Tr I n—underdeveloped trichomes of type I; St n—underdeveloped stomata; and St w—stomata embedded in the epidermis and their guard cells were filled with deposits of epicuticular wax.

**Figure 4 cells-11-03437-f004:**
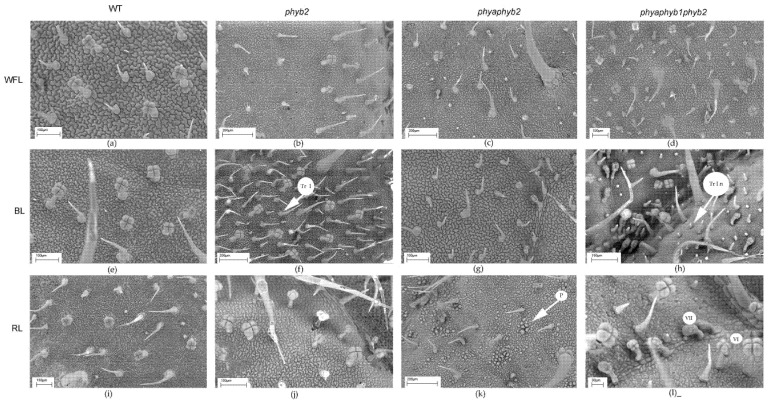
SEM photograph showing the adaxial surface of tomato leaf mutants for the main phytochrome genes: WT (**a**,**e**,**i**); *phyb2* (**b**,**f**,**j**); *phybaphyb2* (**c**,**g**,**k**) and *phyaphyb1phyb2* (**d**,**h**,**l**) under the action of WFL (**a**–**d**), BL (**e**–**h**) and RL (**i**–**l**). Tr I—trichomes of type I; Tr I n—underdeveloped trichomes of type I; VI—trichomes of type VI; VII—trichomes of type VII; and P—papillose formations on the outer surface of the main cells of the epidermis.

**Table 1 cells-11-03437-t001:** Effect of WFL, BL and RL light on photosynthetic rate Pn (µmol CO_2_ m^−2^ s^−1^), transpiration rate Tr (µmol H_2_O m^−2^ s^−1^), Trolox equivalent antioxidant capacity (TEAC), µM Trolox/g FW, gallic acid equivalents (GAE), mg/g FW, and content of flavonoids mg catechin/g FW in the leaves of tomato plants mutant for the genes encoding the main phytochromes. Different letters indicate significant differences (*p* < 0.05) between the experimental treatments.

Variant	Mutants	Pn, µmol CO_2_ m^−2^ s^−1^	T_r_, mmol H_2_O m^−2^ s^−1^	TEAC, µmol Trolox/g FW	GAE, mg/g FW	Flavonoids, mg catechin/g FW
WFL	wt	15.8 ± 1.0 ^a^	0.32 ± 0.02 ^b^	9.17 ± 0.12 ^c^	0.83 ± 0.02 ^c^	0.98 ± 0.02 ^c^
	*phyb2*	10.6 ± 0.6 ^b^	0.12 ± 0.02 ^d^	18.43 ±1.28 ^a^	1.76 ± 0.08 ^a^	1.63 ± 0.10 ^a^
	*phyaphyb2*	8.8 ± 0.4 ^c^	0.45 ± 0.02 ^a^	14.97 ± 1.16 ^b^	1.18 ± 0.10 ^b^	1.32 ± 0.05 ^b^
	*phyaphyb1phyb2*	6.5 ± 0.3 ^d^	0.22 ± 0.03 ^c^	11.08 ± 0.55 ^c^	0.98 ± 0.06 ^bc^	1.14 ± 0.05 ^bc^
BL	wt	11.7 ± 1.0 ^a^	0.48 ± 0.02 ^c^	19.00 ± 0.75 ^c^	1.95 ± 0.20 ^c^	1.99 ± 0.09 ^c^
	*phyb2*	11.2 ± 0.4 ^a^	1.19 ± 0.03 ^a^	50.50 ± 4.24 ^a^	4.95 ± 0.20 ^a^	5.30 ± 0.26 ^a^
	*phyaphyb2*	7.5 ± 0.3 ^b^	0.73 ± 0.04 ^b^	37.37 ± 3.22 ^b^	4.13 ± 0.25 ^b^	4.25 ± 0.30 ^b^
	*phyaphyb1phyb2*	3.1 ± 0.2 ^c^	0.38 ± 0.01 ^d^	15.77 ± 1.74 ^c^	1.27 ± 0.03 ^d^	1.82 ± 0.18 ^c^
RL	wt	14 ± 0.7 ^a^	0.19 ± 0.02 ^c^	9.80 ± 0.46 ^b^	0.98 ± 0.03 ^ab^	0.98 ± 0.08 ^b^
	*phyb2*	9.8 ± 0.6 ^b^	0.48 ± 0.02 ^a^	7.73 ± 1.28 ^b^	0.69 ± 0.16 ^b^	0.79 ± 0.12 ^b^
	*phyaphyb2*	6.7 ± 0.2 ^c^	0.38 ± 0.02 ^b^	13.43 ± 1.39 ^a^	1.30 ± 0.14 ^a^	1.49 ± 0.23 ^a^
	*phyaphyb1phyb2*	3.8 ± 0.2 ^d^	0.16 ± 0.02 ^c^	10.3 ± 0.35 ^ab^	0.63 ± 0.01 ^b^	0.81 ± 0.04 ^b^

**Table 2 cells-11-03437-t002:** Influence of WFL, BL and RL on the main indicators fluorescence chlorophyll Y(II)—PSII effective quantum yield; NPQ—nonphotochemical fluorescence; Y(NO)—quantum yield of nonregulated nonphotochemical energy dissipation in PSII; Y(NPQ)—quantum yield of regulated nonphotochemical energy dissipation in PSII; and F_v_/F_m_—PSII maximum quantum yield in leaves from tomato mutant plants on the genes of major phytochromes. Different letters indicate significant differences (*p* < 0.05) between the experimental treatments.

Variant	Mutants	Y(II)	NPQ	Y(NO)	Y(NPQ)	F_v_/F_m_
WFL	wt	0.411 ± 0.016 ^b^	0.69 ± 0.013 ^b^	0.34 ± 0.006 ^a^	0.24 ± 0.009 ^b^	0.84 ± 0.002 ^a^
	*phyb2*	0.401 ± 0.023 ^b^	1.02 ± 0.111 ^a^	0.29 ± 0.022 ^b^	0.30 ± 0.018 ^a^	0.82 ± 0.009 ^b^
	*phyaphyb2*	0.502 ± 0.013 ^a^	0.89 ± 0.025 ^ab^	0.25 ± 0.004 ^b^	0.22 ± 0.009 ^c^	0.84 ± 0.002 ^a^
	*phyaphyb1phyb2*	0.396 ± 0.029 ^b^	0.83 ± 0.041 ^ab^	0.32 ± 0.008 ^a^	0.27 ± 0.021 ^b^	0.83 ± 0.002 ^b^
BL	wt	0.473 ± 0.008 ^b^	1.10 ± 0.071 ^ab^	0.25 ± 0.009 ^b^	0.27 ± 0.018 ^b^	0.85 ± 0.011 ^a^
	*phyb2*	0.547 ± 0.014 ^a^	0.63 ± 0.058 ^c^	0.27 ± 0.015 ^b^	0.17 ± 0.011 ^c^	0.84 ± 0.004 ^a^
	*phyaphyb2*	0.407 ± 0.009 ^c^	1.16 ± 0.062 ^a^	0.27 ± 0.011 ^b^	0.31 ± 0.009 ^a^	0.84 ± 0.004 ^a^
	*phyaphyb1phyb2*	0.343 ± 0.007 ^d^	0.92 ± 0.024 ^b^	0.34 ± 0.011 ^a^	0.31 ± 0.007 ^a^	0.85 ± 0.005 ^a^
RL	wt	0.374 ± 0.011 ^ab^	0.71 ± 0.071 ^b^	0.36 ± 0.017 ^a^	0.25 ± 0.040 ^b^	0.83 ± 0.002 ^b^
	*phyb2*	0.403 ± 0.007 ^a^	0.66 ± 0.012 ^c^	0.35 ± 0.003 ^a^	0.23 ± 0.003 ^b^	0.82 ± 0.002 ^a^
	*phyaphyb2*	0.413 ± 0.041 ^a^	0.73 ± 0.004 ^b^	0.33 ± 0.024 ^b^	0.24 ± 0.016 ^b^	0.83 ± 0.003 ^b^
	*phyaphyb1phyb2*	0.314 ± 0.014 ^b^	0.93 ± 0.071 ^a^	0.35 ± 0.024 ^a^	0.33 ± 0.038 ^a^	0.79 ± 0.003 ^c^

## Data Availability

The datasets generated during and/or analyzed during the current study are available from the corresponding author on reasonable request.

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

*hy4* mutants grown under blue light. Plant Physiol. Biochem..

