# Peer review of "Effect of Phytochrome Deficiency on Photosynthesis, Light-Related Genes Expression and Flavonoid Accumulation in Solanum lycopersicum under Red and Blue Light"

_cells, 2022, doi:10.3390/cells11213437_

Round 1
Reviewer 1 Report
Using Arabidopsis mutants deficient in Phy activity, the authors examined photosynthesis, flavonoids and phenolic content and antioxidant accumulation under blue and red light exposure. The experiments were designed well and executed efficiently. Except for a few typos, the data is presented coherently and the MS is well written. For researchers engaged in photobiological research, I believe the MS is valuable. It is important that the authors address the following issues, however:
-
Phytochromes and their different types could be briefly discussed by authors, for example, PhyB-PhyE are red photoreceptors that mediate low fluence responses.
-
In other plants, such as rice, Arabidopsis, maize, and so on, there have been reports of PhyB knockouts. Correlate or include the details in your MS.
-
What mutant plant generation was used for the experiment? The T2/T3?
-
Discuss the importance of the results found for the total phenol content and antioxidant capacity.
-
It would be helpful for the readers if there was a hypothetical scheme/diagram summarizing the findings in the discussion. For instance the following finding and other relevant results can be correlated in the form of an illustration:
"We suggest that the increased antioxidant activity under PHYB2 deficiency and the maintenance of high photosynthesis under BL are based on an increase in the expression of the early signalling transcription factors genes BBX, HY5."
Minor points:
-
Although, overall the MS is well-written, I found some inconsistencies, such as Arabidopsis is written as A. thaliana and Arabidopsis throughout the MS.
-
A proofreading should be done for typos: just one example L.511 S.lycopersicumn
Author Response
Dear reviewer,
We would like to thank you for providing helpful suggestions and comments. We have responded to your comments as written below. We hope that you find our revision satisfactory.
- Phytochromes and their different types could be briefly discussed by authors, for example, PhyB-PhyE are red photoreceptors that mediate low fluence responses.
Answer: We made corrections in Introduction section
- In other plants, such as rice, Arabidopsis, maize, and so on, there have been reports of PhyB knockouts. Correlate or include the details in your MS.
Answer: We've added some explanations in the introduction and discussion. It should be noted that, for example, phyb2 mutants are possible only in tomatoes, because the structure of phytochrome genes differs.
- What mutant plant generation was used for the experiment? The T2/T3?
Answer: we used T1 seeds obtained from a seed bank, germinated, and then the plants were cloned.
- Discuss the importance of the results found for the total phenol content and antioxidant capacity.
Answer: We assume that the obtained results can be used to obtain new more resistant tomato varieties for outdoor cultivation. We also assume that it is possible to create conditions for the translocation of flavonoids from leaves into tomato fruits, which will allow creating products with increased nutritional value.
- It would be helpful for the readers if there was a hypothetical scheme/diagram summarizing the findings in the discussion. For instance the following finding and other relevant results can be correlated in the form of an illustration:
"We suggest that the increased antioxidant activity under PHYB2 deficiency and the maintenance of high photosynthesis under BL are based on an increase in the expression of the early signalling transcription factors genes BBX, HY5."
Answer: We tried to draw a model and put it in graphical abstract
- Although, overall the MS is well-written, I found some inconsistencies, such as Arabidopsis is written as A. thaliana and Arabidopsis throughout the MS.
Answer: We made corrections
- A proofreading should be done for typos: just one example L.511 S.lycopersicumn
Answer: We made corrections
Reviewer 2 Report
This research presented the role of various phytochromes in light signalling in tomato plants as well as to elucidate the extent to which this signalling determines the formation of flavonoids and other low-molecular weight antioxidants and affects photosynthesis in tomato leaves. It has been well performed and presented. I only have very minor points that might be addressed.
Introduction:
- Line 58 should be “TFs”
- Line 59-60, 62-63...: Gene names need to be italicized. Please check the full text.
Results:
Major Concerns: -All the results were descriptive and did not provide an in-depth analysis and summary of the role of phytochrome in plant light signals.
- 3.7. Scanning Electron Microscopy: hard to read, The description needs to correspond to the diagram one by one.
- Line 204-206: The English writing severely needs to be improved. Grammatical mistake, please rewrite.
- Line 217: In Table 1, the data shows the content of “gallic acid equivalents (GAE)” , but the author's statement in the section “3.2. Total phenol content” not consisit with the table.
- Line227-228: Unclear to understand, please rewrite.
- Line348-350: There is no description on which figure or table the results are shown. Please added.
- Line353, 365...: “(Figures 2-4)” needs to be clearer
Discussion:
- Line 408-425: Repeat of the results
- Line 452-464: needs to simplification
Author Response
Dear reviewer,
We would like to thank you for providing helpful suggestions and comments. We have responded to your comments as written below. We hope that you find our revision satisfactory.
Responses to your comments:
Reviewer 1
Introduction:
- Line 58 should be “TFs”
Answer: Thank you. We made corrections.
- Line 59-60, 62-63...: Gene names need to be italicized. Please check the full text.
(Проверить весь текст)
Answer: Thank you. We made corrections.
- Results:
All the results were descriptive and did not provide an in-depth analysis and summary of the role of phytochrome in plant light signals.
Answer: We added to Discussion section some points to provide more depth analysis of the role of phytochromes, especislly role of phytochromes in regulation of photosynthetic proteins. Please, see part marked with yellow color.
The observations made here are additionally confirmed by photochemical parameters, since the phyb2 mutant under BL showed the highest effective quantum yield (Y(II)) and the lowest non-photochemical quenching NPQ and Y(NPQ) (Table 2). All this was accompanied by an increase in the expression of the main genes of PSII proteins over the first 24 hours of the experiment (psbA, psbD, CAB6 and psbS), which was not observed in all the other variants (Figure 1). This increase probably indicates the plasticity of the photosynthetic apparatus of the phyb2 mutant, which exceeds the WT values. The phytochrome family of tomato consist of five genes, PHYA, PHYC, and three members of the PHYB subfamily, PHYE and two paralogs of PHYB that have arisen through independent duplication (PHYB and PHYD in A. thaliana, PHYB1 and PHYB2 in tomato). Likely PHYC and PHYE can partly replace participation key PHYA, PHYB1 and PHYB2 in synthesis of phenolic compounds and a difference in content of these compounds between WT and mutants is not so big. In BL role of all phytochromes is decreased and a difference between WT and mutants is getting bigger. Also, in this case expression of genes of photosynthetic proteins D1 (psbA) and D2 (psbD) important for light-induced recovery of PSII and antenna complex gene (CAB6) important for functioning antenna Chl-protein complex was minimum in triple mutant grown under BL and RL, especially at BL. Likely this is one of the main reasons of strong decrease in photosynthetic activity in triple mutant with deficit of key phytochromes (PHYA, PHYB1 and PHYB2). We can see that PHYB1 plays a key role in this decrease. Likely, phytochromes can affect photosynthesis by control of photosynthetic proteins.
- 3.7. Scanning Electron Microscopy: hard to read, the description needs to correspond to the diagram one by one.
Answer: Thank you. We made corrections.
- Line 204-206: The English writing severely needs to be improved. Grammatical mistake, please rewrite.
Answer: We made corrections.
The highest antioxidant activity was observed in the phyb2 and phyaphyb2 mutants under WFL, while in the triple mutant and WT, these indicators were lower and did not differ from each other.
- Line 217: In Table 1, the data shows the content of “gallic acid equivalents (GAE)” , but the author's statement in the section “3.2. Total phenol content” not consisit with the table.
Answer: Section 3.2 title corrected to Total phenolics content. At the same time, there are no contradictions between the terms "Total phenol content" and "Gallic acid equivalents (GAE)". See for example: Food Chemistry, IF=9.231
(https://doi.org/10.1016/j.foodchem.2007.09.046); Analytica Chimica Acta,
IF=6.911 (https://doi.org/10.1016/j.aca.2007.04.037); Journal of Food and Drug
Analysis, IF=6.157 (https://doi.org/10.1016/j.jfda.2013.11.001)
Gallic acid is the accepted standard for expressing content of phenolic compounds in various samples: extracts of various types of plants, wines, juices, etc. As the standard it’s used much more frequently (number of articles in Google Scholar - 53900) than other phenolic acids, for example, caffeic acid (number of articles in Google Scholar - 1520).).
- Line227-228: Unclear to understand, please rewrite.
Answer: We made corrections.
The increase in the flavonoids content in comparison with the control was also observed in the phyb2 (by 65.9%) and double (by 34.1%) mutants under the WFL
- Line348-350: There is no description on which figure or table the results are shown. Please added.
Answer: We made corrections.
- Line353, 365...: “(Figures 2-4)” needs to be clearer
Answer: We made corrections.
- Discussion:
Line 408-425: Repeat of the results
Answer: We made corrections
- Line 452-464: needs to simplification
Answer: We made corrections
Reviewer 3 Report
The results are fantastic. There are several questions about this manuscript.
1. It has been known that the blue light receptor is CRY and red light receptor is phyb. Therefore, it should be talk about in Introduction,
2. Thus, it should be clarified why just choose to study phyb phya, rather than blue light receptor mutant. Why not study blue light receptor mutant simultaneously?
3. The mutants are heterozygosis or homozygous?
4. How could you ensure these mutants just have the mutation of target genes, no other genes?
4. In the Materials, it was said that the plants were exposed to light quality treatment for 7 days. Thus, it is difficult to understand why the leaf appearance like stomata and trichomes number changed in such short time period. The leaf development seems need longer time to change. Thus, it seems impossible to increase or decrease number of stomata and trichomes. please explain this point. Can you find some references to show similar results?
5. The third paragraph in Discussion is so long to follow. Could you split it into several paragraphs? By contrast, there are two paragraphs that are too short, they just have one sentence.
6. Line490, phy1 is phyb1?
7. It is better to draw a model to show the relationship of genes regulation network in different mutants under different light quality.
8. Since the blue light receptor is normally functional in mutants, the changes of low weight molecules, phenols, flavonoids etc. should be discussed about their relationship with blue light receptor. Blue light receptor maybe plays more role than mutated phyA or phyB.
9. The changes in double mutant seems larger than in triple mutants. Why?
10. If phyA and phyB lost function in mutants. Why no significant difference about TEAC, GAE, Falvonoid between WT and mutants under WFL and RL.
Author Response
Dear reviewer,
We would like to thank you for providing helpful suggestions and comments. We have responded to your comments as written below. We hope that you find our revision satisfactory.
Responses to your comments:
- It has been known that the blue light receptor is CRY and red light receptor is phyb. Therefore, it should be talk about in Introduction,
Answer: Thank you. We made corrections.
Phytochromes are among the most characterized photoreceptors. They are RL and FRL sensors and they regulate many developmental processes, including seed germination and hypocotyl growth. Other well-known receptors, the cryptochromes (CRY), perceive light in the blue and UV ranges of the spectrum. They are involved in the growth processes and de-etiolation of seedlings and are also involved in circadian rhythms. Photoreceptors have been studied in detail in Arabidopsis thaliana. They include five phytochromes (PHYA to PHYE) and two major cryptochromes (CRY1 and CRY2)
- Thus, it should be clarified why just choose to study phyb phya, rather than blue light receptor mutant. Why not study blue light receptor mutant simultaneously?
Answer: We used mutants for cryptochromes, and the primary results were presented in the supplementary.
However, how surprising it was, we did not observe significant differences in these mutants. The data are not given in the main text to not overload the article with redundant information.
- The mutants are heterozygosis or homozygous?
Answer: Mutants are homozygous
- How could you ensure these mutants just have the mutation of target genes, no other genes?
Answer: According to the passport, these mutants are monogenic.
Previously in other works, we have shown that mutation at the transcript level corresponds to these mutants. In some cases, there may be a small number of transcripts, but they are greatly reduced. Influence on other genes cannot be ruled out given the exceptional importance of photoreceptor genes in both light and temperature signaling.
Nevertheless, our mutants showed the appropriate phenotypes.
https://onlinelibrary.wiley.com/doi/full/10.1111/pce.13171
https://doi.org/10.1111/nph.17883
- In the Materials, it was said that the plants were exposed to light quality treatment for 7 days. Thus, it is difficult to understand why the leaf appearance like stomata and trichomes number changed in such short time period. The leaf development seems need longer time to change. Thus, it seems impossible to increase or decrease number of stomata and trichomes. please explain this point. Can you find some references to show similar results?
Answer: When a tomato first germinated from seed, it really grows very slowly for the first 4 weeks, and then growth is much accelerated. We have worked with clones that are able to develop very quickly. In 7 days, the plants had time to form several new leaves, and these leaves that were used in the experiments. Another reason why the experiment should run for 1 week is that the triple phytochrome mutant cannot grow under red light conditions for more than 1 week, after which growth slows down. And 2 weeks of red light for a triple mutant is a lethal dose.
Some of our works in which we used similar time intervals:
https://doi.org/10.1016/j.plaphy.2021.07.033
https://doi.org/10.1016/j.jphotobiol.2020.111976
- The third paragraph in Discussion is so long to follow. Could you split it into several paragraphs? By contrast, there are two paragraphs that are too short, they just have one sentence.
Answer: We made corrections.
- Line490, phy1 is phyb1?
Answer: We made corrections.
- It is better to draw a model to show the relationship of genes regulation network in different mutants under different light quality.
Answer: we tried to draw a model and put it in graphical abstract
- Since the blue light receptor is normally functional in mutants, the changes of low weight molecules, phenols, flavonoids etc. should be discussed about their relationship with blue light receptor. Blue light receptor maybe plays more role than mutated phyA or phyB.
Answer: In fact, we also thought so at the beginning of the experiment and planned to obtain slightly different data. Before setting up the main experiments, we performed screening analyzes of a wide range of mutants for different photoreceptors under conditions of red, blue and even green light. In cryptochromic mutants, we observed an increase in the content of flavonoids, however, it was several times lower than in the phyb2 mutant. Preliminary screening studies are attached to the article in Supplementary 2. We concluded that the increase in flavonoid biosynthesis is not so much due to the activation of blue light receptors, how much with a more complete inactivation of phytochromes.
- The changes in double mutant seems larger than in triple mutants. Why?
Answer: We assume that the changes in the double mutant are more noticeable, since phytochrome A, according to which it is mutant, is less important for disrupting the biosynthesis of light-dependent flavonoids under blue light. Together, we observed a change in microphotomorphogenesis and the formation of an outgrowth of epidermal cells in the double mutant, which indicates a possible change in the hormonal status in the double mutant.
- If phyA and phyB lost function in mutants. Why no significant difference about TEAC, GAE, Falvonoid between WT and mutants under WFL and RL.
Answer: The phytochrome family of tomato consist of five genes, PHYA, PHYC, and three members of the PHYB subfamily, PHYE and two paralogs of PHYB that have arisen through independent duplication (PHYB and PHYD in Arabidopsis, PHYB1 and PHYB2 in tomato. Likely PHYC and PHYE can partly replace participation key phytochromes A and B1 and B2 in synthesis of phenolic compounds and a difference in content of these compounds between WT and mutants is not so big. In BL role of all phytochromes is decreased and a difference between WT and mutants is getting bigger. Also, in this case expression of genes of photosynthetic proteins D1 (psbA ) and D2 (psbD) important for recovery of PSII under both BL and RL and antenna complex gene (CAB6) important for functioning antenna complex protein was minimum in triple mutant. Likely this is one of the main reasons of strong decrease in photosynthetic activity in triple mutant with deficit of key phytochromes (PHYA, PHYB1 and PHYB2). We can see that PHYB1 play in this decrease a key role.
In addition, it can be said that there are both flavonoids dependent on the quality of light and independent, which exact isoforms we observe more clearly after the identification and establishment of their chemical structure in our next works.
Changes needed have been added to the text.
Round 2
Reviewer 3 Report
Authors have clarified several puzzles, and the study is interesting. It should be acceptable for publication with minor text editing.
Author Response
We are grateful to the reviewer for careful attention to the manuscript.
We have included corrections in text that has undergone changes during the review process.
The rest of the text was checked by the language editing from American Journal Experts (Certificate verification key 1864-F2E8-BD72-2E2E-F055)